# Flash glucose monitoring in young people with type 1 diabetes—a qualitative study of young people, parents and health professionals: *'It makes life much easier'*

Lucy Beasant [ID],[1] Freyja Cullen,[2] Elizabeth Thomas,[2] Rebecca Kandiyali,[3] Julian P H Shield [ID],[4] David Mcgregor,[5] Nicol West,[6] Jenny Ingram [ID] [1]

For numbered affiliations see end of article.

**Correspondence to**
Dr Jenny Ingram;
jenny.ingram@bristol.ac.uk

## ABSTRACT

**Objectives** Flash glucose monitoring for patients with T1 diabetes avoids frequent painful finger-prick testing, thus potentially improving frequency of glucose self-monitoring. Our study aimed to explore experiences of young people using Freestyle Libre sensors and their parents, and to identify benefits and challenges to National Health Service (NHS) staff of its adoption in their care provision.

**Participants** Young people with T1 diabetes, their parents and healthcare professionals were interviewed between February and December 2021. Participants were recruited via social media and through NHS diabetes clinic staff.

**Design** Semistructured interviews were conducted online and analysed using thematic methods. Staff themes were mapped onto normalisation process theory (NPT) constructs.

**Results** Thirty-four participants were interviewed: 10 young people, 14 parents and 10 healthcare professionals. Young people reported that life was much easier since changing to flash glucose monitoring, increasing confidence and independence to manage their condition. Parents' quality of life improved and they appreciated access to real-time data. Using the NPT concepts to understand how technology was integrated into routine care proved useful; health professionals were very enthusiastic about flash glucose monitoring and coped with the extra data load to facilitate more tailored patient support within and between clinic visits.

**Conclusion** This technology empowers young people and their parents to understand their diabetes adherence more completely; to feel more confident about adjusting their own care between clinic appointments; and provides an improved interactive experience in clinic. Healthcare teams appear committed to delivering improving technologies, acknowledging the challenge for them to assimilate new information required to provide expert advice.

## INTRODUCTION

Type 1 diabetes is a demanding lifelong condition and is the most common form of diabetes in children.[1] Young people and their caregivers need to have the knowledge,

## STRENGTHS AND LIMITATIONS OF THIS STUDY

⇒ This study combines the perspectives of flash Libre from young people of varying ages living with diabetes, their carers and healthcare professionals. The views of less affluent families were heard.
⇒ Patients who had declined to continue flash monitoring are included.
⇒ Recruitment from one region (South-West England) may limit generalisability across the whole country.
⇒ The views of minority ethnic groups are under-represented.

skills and confidence to achieve good glucose control. Young people with poor glucose control have a higher risk of developing long-term complications later in life such as kidney failure requiring dialysis or transplant, limb amputations, blindness and cardiac problems leading to early death.[2]

For many children and young people, management until recently, often required monitoring by finger-prick testing and injection of insulin multiple times a day. Flash monitoring (collecting glucose readings via a device held near to a body-worn sensor) is a relatively new method of monitoring and Abbott FreeStyle Libre (the only flash monitoring device available on the National Health Service (NHS) in the UK) offers an alternative and potentially inexpensive method of glucose monitoring that avoids painful finger-prick testing.[3] The FreeStyle Libre glucose sensor is worn on the arm and measures glucose in the interstitial fluid rather than the blood.[4] While FreeStyle Libre has been welcomed by clinicians and patients, the impact of implementation on healthcare provision has not been studied; one issue with the national roll-out of wearable technologies

is of information overload for patients, their families and clinical teams.

The convenience of Libre compared with finger-prick testing has been shown to increase the frequency of glucose readings, providing families and the clinical team with much more information.[5] It may lead to families becoming more aware of fluctuations which were not picked up by less regular finger-prick testing. This variation can cause concern in parents which needs to be balanced against the comfort, convenience and reassurance that Libre may provide to young people.

Qualitative studies with parents and young people about flash glucose monitoring have described parents' emotional well-being as being improved and their role in diabetes management facilitated.[6] Young people have reported that flash monitoring reduced management burden and improved glucose control.[7] In a recent paediatric trial of 'Hybrid' closed-loop technology using the Dexcom continuous glucose monitoring device (with an app and insulin pump), parents reported multiple benefits to healthcare professionals being able to access their child's glucose/insulin data remotely, leading to improved consultations, better clinical input and support from healthcare professionals between consultations.[8]

The FLASH study (implementation of flash glucose monitoring in four paediatric diabetes clinics—a controlled before and after study to produce real world evidence of patient benefit) aimed to explore (1) whether flash glucose monitoring improved outcomes in children with type 1 diabetes compared with finger-prick testing; and (2) what the healthcare implications associated with flash monitoring, when compared with finger-prick testing, were in terms of staff time and healthcare costs.[9]

Alongside the before and after study, our qualitative study aims were (1) to provide feedback on using Libre with this patient group by exploring the views and experiences of children, young people and their parents, in the form of co-produced resources; and (2) to identify the benefits and challenges to NHS staff of using Libre in their care provision.

## METHODS

Qualitative interviews with young people included children and young people (aged 8–18) with type 1 diabetes who currently use FreeStyle Libre and/or finger-prick testing, have stopped using Libre, or chose not to take up Libre. Parent/carer (hereafter 'parent') interviews included parents of children aged 5–18 years.

Families were recruited using two routes: volunteers via social media (Facebook, Twitter, Diabetes UK online notice boards); and patients from one NHS diabetes centre. Participants were purposively sampled to include a range of ages of child, time from diagnosis and socio-economic deprivation.

Staff included in the qualitative focus group and interviews were those in direct clinical care of children/young people with type 1 diabetes. Purposive sampling was used

to include a range of healthcare roles across NHS sites in South-West England.

Normalisation process theory (NPT) was used to understand the integration of Libre into routine healthcare delivery. NPT proposes that implementation of complex interventions is dependent on the ability of participants to fulfil four criteria (the core constructs of NPT). These are 'coherence' (how people make sense of the intervention), 'cognitive participation' (the work people do to develop new practices), 'collective action' (the work to operationalise practices), and 'reflexive monitoring' (ways in which people appraise how new practices are working).[10]

All the interviews were conducted online or by telephone during the COVID-19 pandemic (February–December 2021).

### Data collection
#### Interviews with children, young people and parents
Parents or young people (including social media volunteers) were asked to complete a JISC eligibility survey (a UK General Data Protection Regulation (GDPR) compliant platform https://www.onlinesurveys.ac.uk/help-support/online-surveys-security). This survey collected demographic information, diabetes duration and methods of glucose measurement. From November 2021, we obtained assent/consent to contact from families through one paediatric diabetes centre in SW England which held lists of eligible patients. Paediatric diabetes nurses asked families for consent to further contact about the FLASH study. Contact details were sent to the research team (via NHS email) who emailed families the relevant information sheets and consent forms.

Young people were interviewed with or without their parent depending on their age and preference. The interviews were conducted by an experienced qualitative researcher (LB) and explored their experiences of using Libre and factors which influence uptake and continued usage (or not). We also included some young people who stopped using Libre and those who used Libre for varied periods of time. The interview topic guide focused on flash monitoring experiences, self-efficacy and confidence in using Libre (see online supplemental material).

#### Staff focus group and interviews
South-West England healthcare professionals were invited to participate in a focus group run by LB and JI; those unable to attend were interviewed at a convenient time (by LB). Interviews/focus group explored their views and experiences of the Freestyle Libre monitor, and the topic guide (see online supplemental material) was developed using the core constructs of NPT.[10]

### Data analysis
All interviews were transcribed and anonymised, read and coded by research team members with backgrounds in psychology and health service research. Analysis began shortly after data collection started and was an ongoing

iterative process, drawing on a data-driven inductive thematic analysis approach.[11]

NVivo V.12 software was used to organise and code the transcripts (QSR International Pty Ltd). Two researchers coded a sample of transcripts independently and compared the coding; any discrepancies were discussed within the research team and resolved to achieve a coding consensus and ensure robust analysis. Themes from the health professional data were mapped on to NPT constructs[10] and those developed from the family interviews, were compared, and contrasted with the staff themes. All data were analysed in batches, and sampling continued until no new themes or constructs were developed from the data.

### Patient and public involvement to co-produce resources

We planned to hold some in-person, co-production workshops to design resources for young people and parents. Due to the COVID-19 pandemic lockdowns the discussions to produce animations for young people aged 8–12 years and 13–17 years and infographics were held online. We held two Zoom groups with families and teenagers to prioritise what was needed and invited the young people and wider PPI group members to narrate the animations using direct quotes from our findings.

Ethical approval was granted by the University of Bristol REC on 26 January 2021 for social media recruitment for interviews and by South Birmingham REC (IRAS 286988) on 20 October 2021 for NHS recruitment.

## RESULTS
### Family demographics

Fourteen families were included as shown in table 1, and interviews took place by Zoom or telephone; average interview length was 43 min. Eight young people were interviewed from the 12 core families and a further two teenagers from different families were interviewed without their parents. Seven young people were interviewed with their parent(s) present and three were interviewed individually. Nine families were based in the South-West, two West-Midlands, two South-East and one in London. Five families were from areas classed as least deprived (Index of Multiple Deprivation (IMD) 8–10), four lived in areas of average deprivation (IMD 4–7), four lived in most deprived areas (IMD 1–3) and we had no postcode for one family. Eight young people were using Libre 2, four were using Libre 1, one had stopped using Libre 1 and another child not interviewed had switched to a continuous blood glucose monitor (CGM).

### Health professionals' demographics

Ten health professionals took part in interviews (n=5) or a focus group (n=5) which were conducted by Zoom or telephone. The average interview length was 44 min and the focus group lasted 57 min. Five paediatric diabetes specialist nurses, four consultants and one allied health professional participated. Seven of the health professionals were females and three were males and they were based in seven hospitals across South-West England.

### Young people and parent themes

Four themes were developed from the data to describe the views and experiences of the young people and their parents as shown in table 2. The first three themes ('It makes life much easier', 'challenges and difficulties' and 'confidence and independence') were identified for both groups and the fourth ('the tech journey') was particularly discussed by the parents. The themes are illustrated by quotes in table 2 with individuals identified by an anonymised participant number, child's age group (<8 years, 8–12 years, 13–17 years) and whether they were currently using Libre 1 or 2.

#### It makes life much easier (children and young people)

This theme describes the triggers and perceived benefits of using Libre. Young people discussed their school, work and social life being 'easier' since they started using Libre. The ability to check blood glucose levels quickly via scanning, without stopping what they were doing was highly valued. They reported that they could not 'feel' the Libre sensor when it was on their arm. Libre was described as more convenient than finger-pricking because it needed less equipment and most importantly, because it significantly reduced the physical impact on fingers due to frequent finger-pricking (quote 2.1.1). Participants also reported benefit at night such as being woken (by the Libre 2 alarm) and being able to treat a hypoglycaemic episode easily and sooner which resulted in them not feeling so terrible when they woke in the morning. Libre 2 allowed them to carry on with activities including sport at school or paid work using the alarm or vibrate function to alert them to out of range glucose levels (quote 2.1.2).

#### Living their best possible life (parents)

Parents reported the huge benefits of using Libre. It was apparent that Libre not only made their child's life

| Table 1 | Demographics of young people and parents interviewed | | |
|---|---|---|---|
| **Children and young people interviewed** | **n=10** | **Parents interviewed** | **n=14 from 12 families** |
| Age range | 8–17 (average 14 years) | Child's age range | 6–17 (average 12 years) |
| Gender | 5 Female, 4 males, 1 prefer not to say | Parent gender | 11 Mothers/female carers, 3 fathers |
| Ethnic group | 9 White, 1 white/black Caribbean | Ethnic group | 14 White |

**Table 2** Interview quotes from young people (n=10) and parents (n=14) organised by theme

| Theme | Quotes—anonymised, with child's age group and Libre1/2 |
|---|---|
| 1. It makes life much easier—Children and young people<br><br>Living their best possible life— Parents | 2.1.1 *'It's quite cool that we now have this technology that* **makes my life a lot easier***. I just find it so much easier to have something to scan… I feel like I would rather just have one prick in my arm once every 14 days than have to do it 14 times a day. That's one of the main reasons why I like It'.* **#115, 13–17 years, Libre 2**<br>2.1.2 *'I just check my watch… it's really helpful because it can show where I have highs or lows every day'.* **#106, 8–12 years, Libre 1**<br>2.1.3 *'great for me for peace of mind… It makes my life ten times easier because you think oh yeah 6.7'.* **#104 parent, child 8–12 years, Libre 1**<br>2.1.4 *'It's great for swimming and things like that, that you can just get in the pool, get out, scan, you know where you are'* **#103 parent, child 8–12 years, Libre 2** |
| 2. Challenges and difficulties | 2.2.1 *'It was a bit of a shock when my Libre could say I was high like 16 or something, but really I would be 11 on the blood readings. So there was a little bit of the inaccuracy once you hit a certain number was a little bit annoying, but most of the time it was pretty good'.* **#119, 13–17 years, Libre 2**<br>2.2.2 *'I didn't really pay attention as such to the videos because it was covering stuff we already knew. They were just painfully long, they were 20 minutes each video and there five of them you had to watch'.* **#108, 13–17 years, Libre 1**<br>2.2.3 *'I do get quite a lot of comments saying, "What is that?" And then I have to explain it to them'.* **#106, 8–12 years, Libre 1**<br>2.2.4 *'We got to a point where I think we were just getting overwhelmed with all the information, the 24 hours of the information'.* **#108parent, child 13–17 years, Libre 1**<br>2.2.5 *'Occasionally she will have two on where we can't get one off… we have to use a peel spray'.* **#102 parent, child <8 years, Libre 2**<br>2.2.6 *'From a carer's point of view they (alarms) are reassuring, from a younger child point of view they are incredibly annoying'.* **#103 parent, child 8–12 years, Libre 2** |
| 3. Confidence, independence and more effective monitoring. | 2.3.1 *'I know it's not going to come off because every time you try and get it off like at the end of the 2 weeks it was really hard. So I am confident in knowing it's not going to come off'.* **#105, 8–12 years, Libre 2**<br>2.3.2 *'I quite like people knowing, or seeing it… It means other people that are diabetic know I'm diabetic, which I think is quite cool when you're in the street and you see someone else with one'.* **#101, 13–17 years, Libre 2**<br>2.3.3 *'I think it's opened up the doors for sport… he will go for a run now with the scanner and some glucose and we know that he will be safe because he will act on it'.* **#108 parent, child 13–17 years, Libre 1**<br>2.3.4 *'When we're making changes to ratios and insulin dosage and everything then the graphs are quite good for that to see trends in everyday, and then through weeks and months and everything, and then that can help pinpoint it'.* **#119, 13–17 years, Libre 2**<br>2.3.5 *'It was better on both sides, he knew what he was doing in the daytime, he knew he could go into PE and be fully confident, likewise we were happy with the night-time'.* **#107 parent, child 8–12 years, Libre 2** |
| 4. The 'tech journey' | 2.4.1 *'Because he's a 12-year old boy relying on him to scan regularly is hit and miss, so at least getting the(xx add-on)it took away the need for him to scan. He still has to scan every eight hours otherwise the hospital can't see his data, but I can look on my watch and I can see what his blood sugars are at any time of the day and in the night, which I do, that's what we use it for really'.* **#106 parent, child 8–12 years, Libre 1**<br>2.4.2 *'It would be good to have training for when there's been something wrong with my blood sugar, and maybe discussing changing ratios and stuff, and just maybe just talking about things that are stressing me out'.* **#105, 8–12 years, Libre 2**<br>2.4.3 *'It becomes one of your everyday tools, and essentially becomes a bit of a part of you'.* **#119, 13–17 years, Libre 2** |

better by giving them more freedom to live life to the full, despite their diabetes diagnosis, it also improved parents' quality of life by reducing the constant anxiety (quote 2.1.3). Parents felt that Libre was more convenient than finger-pricking when out and about, simple to use, with benefit at school and when out socially with friends. They felt that their child also missed fewer activities at school and helped them stay safe for sports (quote 2.1.4).

## Challenges and difficulties

Considering the barriers to using flash monitoring, there were several challenges and difficulties experienced by both young people and reported by parents. These included technical issues such as faulty sensors, reduced accuracy when blood glucose was very high or low and a lack of trust in Libre when adjusting to it in early days of use (quote 2.2.1). The signal loss

alarm sounds the same as the high/low alarm (Libre 2), which was stressful for young people. Some families commented that they needed more tailored advice, support and training, particularly when their child was newly diagnosed (quote 2.2.2).

For some young people (particularly those aged 8–12) there was still the issue of the sensor being visible on their arm and questions from others caused them to hide it under clothes by wearing long sleeves. However, many were able to explain what it was for and answer these questions. Others still felt self-conscious, particularly if jokes were made about being 'scanned' or when friends 'tried to scan me', which felt like an invasion of privacy (quote 2.2.3). Finger-pricking enabled a child not to think about their diabetes as much, but using Libre and having access to continuous data could be overwhelming or demoralising particularly when their management was poor (quote 2.2.4). Parents reported varied experiences in relation to adhesion of the sensor, some had issues with Libre staying attached in hot weather, while others found it hard to remove, which was also challenging (quote 2.2.5). Some parents reported more stress and anxiety with the increase in the quantity and availability of data. Conflicts between parents and children occurred when they were woken by the alarms at night when they were tired, or when their blood glucose was rising but they wanted a snack so a correction would be needed (quote 2.2.6).

### Confidence, independence and more effective monitoring

Young people's self-efficacy was increased as they felt more confident going about their daily lives when wearing their Libre sensor, at school, work and socially. Some felt that Libre made their diabetes less visible—scanning the sensor on the upper arm draws less attention when they needed to check their levels at school or in public. This was also helped by the fact that Libre does not fall off and can be adapted (using vibrate, link to watch) to fit in with their routine, so they can be independent throughout the day (quote 2.3.1). This reduced visibility was particularly important for younger teenagers and those who felt self-conscious or different because of their diabetes. In contrast, others liked their sensor being visible, they felt more confident because others might be more likely to act should they have a hypoglycaemic episode in public. Two older teenagers liked the visibility of Libre because others with diabetes, who also use Libre, know what it is, and they can acknowledge each other (quote 2.3.2).

Young people and their parents described that having real-time access to their own data made them able to make more informed decisions. Young people were able to build confidence in managing their condition effectively and have more independence from their parents and clinical team by, for example, being able to go for a run on their own or adjust insulin ratios at home using data from their device (quote 2.3.3).

Libre data output features such as glucose prediction arrows were particularly useful, as was the alarm or vibrate feature in Libre 2. Some used the graphs and charts which allowed them to make adjustments and predictions, and plan ahead to develop a scanning routine. Parents liked the fact that they could scan their child at night with Libre 1 or use the alarm with Libre 2. They could also monitor from afar as they could see their child's data on their phone when the child scanned at school. Data were also used to make reflective decisions with their clinical team (quotes 2.3.4, 2.3.5).

### The 'tech journey'

This final theme was mainly discussed by parents who described their child's journey from finger-pricking many times a day to scanning using Libre and Libre 2. Parents were always on the lookout and striving for better technology and combinations of technology to improve management. However, they also needed to be engaged in the progress of diabetes technology and be 'tech-savvy' to set it up and make third-party add-ons (to turn the flash monitoring into a continuous glucose monitoring system) work as these are not supported by Abbott or the NHS (quote 2.4.1). This also has the potential of widening health inequalities, as those on lower incomes or with restricted internet access are disadvantaged. Self-funding the Libre was an issue for some early adopters of Libre; one family stopped using it due to cost, then started again when it was funded via the NHS.

Generic training as a 'condition of use' was skipped by some families who felt they already knew what they were doing, so tailored training support may be more appropriate for many and particularly something suitable for young people would be helpful (quote 2.4.2). However, the cost implications of such tailored training should be acknowledged.

Overall young people wanted to keep using flash monitoring sensors and parents felt positive about the future for their child because technology (including CGM and insulin pumps) was always evolving and getting better (quote 2.4.3).

### Co-produced animations

The animations were developed with young people providing the narration and several reading their own quotes: *'It makes life a lot easier': 8–12 year-olds on 'flash' glucose monitoring:* https://youtu.be/yORUIplJeBs; *'Feeling confident and independent': 13–17 year-olds on 'flash' glucose monitoring:* https://youtu.be/M2tw093IcCw.

We also produced an infographic for parents to be disseminated with the animation links. These are hosted on Digibete (digibete.org) and the NIHR Bristol Biomedical Research Centre website: https://www.bristolbrc.nihr.ac.uk/FLASH-animations .

### Health professional themes

Four themes with several subthemes were developed from the health professional interviews and focus group and mapped to the NPT constructs. These and the quotes are

**Table 3** Health professional interview quotes mapped to the constructs of normalisation process theory

| NPT construct and theme | Quotes—anonymised clinicians |
|---|---|
| 1. Coherence—adapting to change and making sense of the technology. | 3.1.1. *If our IT systems aren't working the clinic is a nightmare, so you can't look at these things, and understandably families get irritated. So there is a constant IT challenge, but on the whole it's better than what it was.* **#208 Consultant**<br>3.1.2. *We all agree that we can't keep up with the technology and that the intricacies of how each thing works and the details of what this one can do and what that, we just can't keep up with it, because it is forever changing.* **#202 PDSN**<br>3.1.3. *Some kids like that and others they don't want to be constantly reminded, and they don't want their parents texting them when they are out and about with their friends saying your blood sugars are low, your blood sugars are high what are you doing?* **#208 Consultant** |
| 2. Cognitive participation—embracing new technology and buy-in | 3.2.1. *'We run download clinics where we look at downloads….it's really good, and actually to be honest it's quite hard to look at a download now without Libre or xx, because it does fill in those gaps'.* **#205 PDSN**<br>3.2.2 *'I think we work through the challenges as they come, but we are trying to just make it work for each family because we believe in Libre, that it's beneficial…'* **#206 AHP** |
| 3. Collective action—better monitoring, tailored support, working to operationalise practices | 3.3.1. *'it's just such a visual picture for them to understand what is actually happening, and because it's your own body, it's not just textbook, I think it really resonates with them into action'.* **#206 AHP**<br>3.3.2. *'it becomes a useful tool for managing diabetes rather than just watching your diabetes'.* **#224 Consultant** |
| 4. Reflexive monitoring—looking to the future and appraisal of new practices | 3.4.1. *'I have noticed that they (Abbott) are altering the training as they go along so it's more fit for purpose… they have made it a bit more specific… I think parents are going away with a bit more understanding about how they are actually going to use this bit of kit'.* **#225 PDSN**<br>3.4.2. *'With our people who haven't got mobile phones or who are of low income, the whole technology thing is stacked against them I think in some ways because a lot of this further increases the health inequalities'.* **#221 Consultant**<br>3.4.3. *'I am not sure how comfortable teachers are at looking at it, but the young people themselves will know if things were okay or not'.* **#208 Consultant** |

AHP, Allied Health Professional; NPT, normalisation process theory; PDSN, paediatric diabetes specialist nurse.

shown in table 3, with health professional identified by role and anonymous ID.

### Coherence—adapting to change and making sense of the technology

Staff described the steep learning curve and time involved in understanding the software used and large amount of data from not only Freestyle Libre patient sensors but other new systems of CGM. Some families wanted to go through every detail of the output and often there was not enough time in clinic appointments to do this. There were difficulties with IT systems including having to open several different applications, understanding Libre-specific issues, and trying to download data in the clinic (quotes 3.1.1, 3.1.2). Staff felt that the volume of data and technology issues made many parents more anxious, whereas some young people did not want to be reminded of their diabetes via the constant stream of data, as reported in 'challenges and difficulties' (quote 3.1.3).

### Cognitive participation—embracing new technology and buy-in

Health professionals reported that they were highly committed to Libre, used it to educate families and were pleased to see improvements with the newer models of Libre which had made things more accurate and easier to use (quote 3.2.1). They were also keen to know about the effectiveness of Libre on glycated haemoglobin levels and diabetes management from the FLASH quantitative study. They mentioned the positive impacts for families, both psychological, particularly for parents at night, and the physical improvements for patients with less finger-pricking (quote 3.2.2). These were also important points highlighted in the 'making life much easier' theme for young people and parents.

### Collective action—better monitoring, tailored support, working to operationalise practices

Staff highlighted that Libre enabled them to improve management of patients' diabetes between clinic visits and use it as a tool for managing rather than just watching their diabetes. They could also tailor their support in clinic appointments as they had the glucose monitoring picture and so could make informed clinical decisions with families. It is also easier to see and understand the impact of activities or meals and so clinic staff were able to give positive feedback in clinic about good management. When diabetes management was poor, ineffective monitoring was more visible with Libre (quotes 3.3.1, 3.3.2). These points were highlighted by parents and teenagers as 'helping them feel more confident and independent'.

### Reflexive monitoring—looking to the future and appraisal of new practices

When reflecting on the introduction of flash glucose monitoring devices, health professionals highlighted some interesting points. Using online training has become more acceptable since the COVID-19 pandemic lockdowns, however they felt that the webinars are still largely 'adult centric' and not very suitable for young people (quote 3.4.1). Many clinic appointments were held online during the pandemic and having the ability to monitor glucose levels remotely made the process more efficient and effective.

Using new technology can widen health inequalities for those who do not have up to date smartphones or no access to wi-fi as they cannot use Libre as effectively (quote 3.4.2). Staff suggested creative ways to overcome these inequalities such as identifying a mobile phone as a medical device in the care plan to allow it to be prescribed and using charities to supply phones for those with limited access.

They also discussed whether training for teachers who had young people with type 1 diabetes in their classes was appropriate and feasible (quote 3.4.3).

## DISCUSSION

Our study reported on 34 interviews with young people, parents and health professionals about the facilitators and barriers to using flash glucose monitoring for young people with type 1 diabetes. Life was much easier and better for young people freed from the burden of frequent finger-pricking. Changing to flash glucose monitoring gave them confidence and increased independence in managing their condition. Parents' quality of life also improved, and they appreciated the access to real-time data. Conflicts between parents and children occurred but were largely overcome, with the majority of young people still choosing to use Libre because it resulted in them having to finger-prick less often. Challenges with IT and increased visibility were mostly successfully overcome by this group of young people and parents. Mapping our data onto the NPT concepts to understand how staff integrated Libre into routine care especially during a pandemic was a helpful approach.[10] Health professionals reported that they were very enthusiastic about flash glucose monitoring, felt they were coping with the extra burden on their time and data overload and were keen to continue using new and improved Libre models in future. It helped them give more tailored support within and between clinic visits, especially when most of their contact with families was online. However, they were concerned about the potential widening of health inequalities for those with poor internet connections and suggested ways around some of the problems. Better education resources for young people and teaching staff were also suggested as being helpful. We hope that our two short animations for children and teenagers which are narrated by young

people and used direct quotes from our findings will help to provide more accessible information for them.

Other studies reporting on flash glucose monitoring have highlighted the positive aspects of improved monitoring and quality of life for young people and parents but also some sensor-related challenges when using them initially including poor adhesion reported in one survey.[3 5 6 12] Our findings concur with these recently published studies, but importantly we have added to the available knowledge base by recruiting those who had recently started, stopped or had been using Libre for some time in England. Our novel finding relating to the 'visibility' of Libre and the potential for young people to realise the potential for positive acknowledgement from others, or intervention from the wider public should the wearer have a hypoglycaemic episode, add to existing evidence in this area that reports visibility to others as a moderately negative feature of the device.[6] Parents involved in our study were often very 'tech-savvy', and the 'tech journey' they described highlights the potential for further health inequalities as diabetes technology quickly changes, resulting in families on low incomes not being able to access new technology until it is funded on the NHS.

Flash glucose monitoring and other CGM devices have the potential to improve long-term glycaemic control.[13 14] Young people should be supported and encouraged to use the sensors effectively.[7 13 14] Our findings suggest that manufacturers should provide tailored education that is more accessible to children and young people, and clinicians should continue to support families to make informed treatment decisions to maximise the benefits of this technology. This was shown in a recent paediatric trial of continuous glucose monitoring devices (not involving Freestyle Libre) across four European countries, which reported improved consultations and better clinical input and support from healthcare professionals between consultations.[8]

### Limitations

Recruitment via online forums and patients predominantly from one area within South-West England will limit the generalisability of the findings to those who were engaged with accessing information online. We were only able to interview patients and parents of mostly white ethnicity which will also have reduced the spectrum of generalisability. However, we did manage to include individuals from less affluent populations and a few who had stopped using flash monitoring.

*Implications for practice* include observations that using Freestyle Libre sensors improves the consultation process as visible data on a clinic computer allow collective visualisation of good and less good control and a collaborative process of adjustments to be made which facilitates greater patient/family involvement in their care.

The ability to view data at distance hugely improves web-based consultations meaning that eventually, some but not all clinic appointments could be held remotely.

If COVID-19 or similar comes back, it will be a significant tool in maintaining good care as it did in the COVID-19 pandemic. The socioeconomic and education disparities in access and understanding are still to be fully addressed. A concerted effort by the local paediatric diabetes team has improved uptake by ethnic minority groups, but with a lag compared with early adopters who tended to be more advantaged and better informed.

## CONCLUSIONS

This technology empowers both young people and their parents to understand their diabetes adherence more completely; to feel more confident about adjusting their own care between clinic appointments; and provide an improved healthcare professional-young person/parent interactive experience in clinic. Health professionals were positive about its use, reporting practical, clinical and educational benefits for patients, while also identifying the increased time commitments for individual family support within limited healthcare resources. With the rapid adoption of some form of CGM for all children with diabetes and evolving intricacies of insulin delivery, it may be the case that reviewing downloads regularly and the additional time to analyse at each clinic visit, may eventually reduce the number of face-to-face appointments available for each child and family due to limited staffing resources.

**Author affiliations**
[1]Centre for Academic Child Health, Bristol Medical School, University of Bristol, Bristol, UK
[2]Children's Diabetes Support, University Hospitals Bristol & Weston NHS Foundation Trust, Bristol, UK
[3]Centre for Health Economics, Warwick Medical School, Coventry, UK
[4]Translational Health Sciences, Bristol Medical School, University of Bristol, Bristol, UK
[5]Royal Devon and Exeter NHS Foundation Trust, Exeter, UK
[6]Great Western Hospitals NHS Foundation Trust, Swindon, UK

**Acknowledgements**  We would like to thank all the young people, parents and health professionals involved in the interviews, the PPI group, and those who helped make the animations. The views expressed here are those of the authors and not necessarily those of the NHS, the NIHR or the Department of Health and Social Care.

**Contributors**  JI, LB, RK and JPHS made substantial contributions to conception and design of the study. LB, ET, FC, DM and NW were involved in acquisition of data. LB and JI were involved in the analysis and interpretation of data. All authors were involved in drafting the manuscript and revising it critically for important intellectual content and have given final approval of the version to be published. JI acts as the guarantor for the qualitative study.

**Funding**  This research is funded by the National Institute for Health Research (NIHR) Research for Patient Benefit Programme (#NIHR201085). The views expressed are those of the authors and not necessarily those of the NIHR or the Department of Health and Social Care.

**Competing interests**  None declared.

**Patient and public involvement**  Patients and/or the public were involved in the design, or conduct, or reporting or dissemination plans of this research. Refer to the Methods section for further details.

**Patient consent for publication**  Not applicable.

**Ethics approval**  Ethical approval was granted by the University of Bristol REC on 26 January 2021 for social media recruitment for interviews and by South

Birmingham REC (IRAS 286988) on 20 October 2021 for NHS recruitment. Participants gave informed consent to participate in the study before taking part.

**Provenance and peer review**  Not commissioned; externally peer reviewed.

**Data availability statement**  Data are available upon reasonable request. Anonymised data are available on reasonable request from the corresponding author via the University of Bristol Freedom of Information office.

**ORCID iDs**
Lucy Beasant http://orcid.org/0000-0002-4279-5644
Julian P H Shield http://orcid.org/0000-0003-2601-7575
Jenny Ingram http://orcid.org/0000-0003-2366-008X

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
