## [Reviewer comments · BMJ Open]

ARTICLE DETAILS

TITLE (PROVISIONAL)	Flash glucose monitoring in young people with type 1 diabetes, a qualitative study of young people, parents, and health professionals: 'It makes life much easier'.
AUTHORS	Beasant, Lucy; Cullen, Freyja; Thomas, Elizabeth; Kandiyali, Rebecca; Shield, Julian; Mcgregor, David; West, Nicol; Ingram, Jenny

VERSION 1 – REVIEW

REVIEWER	Urakami, Tatsuhiko Nihon University School of Medicine Graduate School of Medicine
REVIEW RETURNED	25-Dec-2022

GENERAL COMMENTS	Manuscript ID bmjopen-2022-070477 entitled "Flash glucose monitoring in young people with type 1 diabetes, a qualitative study of young people, parents, and health professionals: 'It makes life much easier'." The authors described that Flash glucose monitoring (FGM) enabled young people with type 1 diabetes and their parents to understand their diabetes adherence more completely and to feel more confident about adjusting their own care by analyzing the interviews from them. This article is of clinical interest, however there are major problems for publication in BMJ. First, the participants of interview were few, and I consider the analyses of the results were incomplete to draw the conclusion: i.e., this study included only ten young people, and their parents and caregivers. The age-range in the participants was 8-18 years old, but young children aged less than 7 years were not studied, who may be the most important age-group for evaluation of their diabetes care and their QOL. CGM is more useful for young people in the management of diabetes. Second, the results may be different between the age-groups. Therefore, the results should be compared between the age-groups, 8-12 year- and 13-18 year-group. However, I understand the comparison seems difficult because the number of patients in each group was too small for the statistical analysis. Third, the patient background was insufficiently described. Age at diagnosis, duration of diabetes, insulin treatment (MDI or CSII), HbA1c level, and serum C-peptide level in the patients should be described. Fourth, there is some difference between Libre 1 and Libre 2: i.e., Libre 2 has more benefit such as having hypo-/hyperglycemia alert system. The accuracy, particularly in low glucose level, is superior in Libre 2. Therefore, there may be difference of the results between the users with Libre 1 and those with Libre 2. The results
--

	in the study may be different according to use of Libre 1 and that with Libre 2. Fifth, interpretation of the results was complicated in the study. The authors should summarize the results easy to understand. Sixth, the usefulness of real-time CGM for children with type 1 diabetes should be mentioned in the discussion. Finally, the references lack important studies described the clinical use of FGM in children and adolescents with type 1 diabetes. The ISPAD Consensus Guideline 2022 is at least necessary for the discussion. Recent other studies should be included in the references.
--	---

REVIEWER	ANDREELLI, FABRIZIO Hôpital Pitié-Salpêtrière Service Néphrologie
REVIEW RETURNED	04-Feb-2023

GENERAL COMMENTS	The study of Lucy Beasant and colleagues aimed “to analyze experiences of young people using Freestyle Libre sensors and their parents, and to identify benefits and challenges to NHS staff of its adoption in their care provision”. Young people with T1 diabetes, their parents and healthcare professionals were interviewed between February and December 2021. After anonymization, all interviews were transcribed, read and coded. As usual, NVivo 12 software was used and a coding consensus was achieved among experts. Interviews were analyzed until no new themes and constructs were developed from the data. Using this classical methodology in the field of social sciences, the number of participants is usually small and depends of the number of the themes explored in the study. Table 2 showed some interview quotes from young people and parents organized in 4 themes: ‘It makes life much easier’, ‘challenges and difficulties’, ‘confidence and independence’ were identified for both groups and the fourth (‘the tech journey’) was identified for the parents. Table 3 showed the health professional interview quotes organized in 4 themes :” Coherence - Adapting to change and making sense of the technology”, “Cognitive participation - Embracing new technology and buy-in”, “Collective action – Better monitoring, tailored support, working to operationalise practices” and “Reflexive monitoring -Looking to the future and appraisal of new practices” The methodology is well described and appropriate for the present study and reached the high level of quality in research in social sciences (or social psychology). There is no ethical issue. Major concerns 1-The description of the results (on the one hand for the patients and their parents and on the other hand for the healthcare professionals) summarized adequately the most important messages from the interviews and also the constructs from the health professional data. But Freestyle Libre sensors are used since 2017 in many countries in Europe. Individual interviews and focus groups in patients with type 1 diabetes and their parents in our routine clinical care, confirmed all the data presented in the
--

	paper of Lucy Beasant and colleagues especially (some interesting sentences from the discussion)  - Life was much easier and better for young people freed from the burden of frequent finger-pricking - Changing to flash glucose monitoring gave them confidence and increased independence in managing their condition - Parents' quality of life also improved, and they appreciated the access to real time data. -Health professionals reported that they were very enthusiastic about flash glucose monitoring and were coping with the extra burden on their time and data overload. Similar conclusion has already been published since 2017 and the references of these papers are listed in the manuscript of Lucy Beasant and colleagues. Thus, the discussion did not provide comparison with the literature and failed to demonstrate sufficient novelty in this field. 2- In the discussion, the following sentence "Manufacturers and clinicians should provide good education on interpreting glucose data and support making informed treatment decisions to maximize the benefits of this technology" is strange because the use of Free Style libre sensors in adults and in children with type 1 diabetes is a part of our routine care including the analysis of the data by the patients and their parents. This sentence suggested that this sensor is a new device that is currently being evaluated. And this is not the case. 3-Beyond the initial analysis of the data, the similarity or difference of the themes and constructs addressed by each category of participants (patients, parents, health care professionals) should be the most interesting conclusion of such study. The discussion does not sufficiently highlight the differences in participants' expectations. For example, some children and adults refuse to use this sensor or disconnect the alarms despite the improvement of long-term glycemic control established in clinical studies. Their decision is in contrast with the expectations of their parents. How can we explain these differences? How each category considers type 1 diabetes mellitus: a jail for young patients ? A challenge for parents to succeed ? A new data collection like the "temperature curve" or "equivalent to accumulated biological data over time such as kidney or liver function, needing mathematical analysis with means and data distribution but far away the specific history of each patient for health care professionals"? Can we develop some specific strategy to convince the patients to use these sensors? In this strategy of permanent glucose monitoring, to what extent parents and professionals may accept mistakes of patients in their routine management of the disease ? On the contrary, thinking that everybody can see your mistakes may be stressful for the patient community. Is it a sufficient reason to reject the use of the device? Yes, if we consider that parents and children share the same data in real life: less stress for parents; more discussion and sometimes conflicts for the young patients with their parents. This sentence of the discussion "Young people should be supported and encouraged to use the sensors effectively" is key and should be developed considering the data available in the literature and also the analysis of the limitations mentioned by the patients. In other words, such device has first to help patients and the discussion seems less patients-centered than one could expect. In addition, as mentioned above, the results of the present study
--	---

	sometimes did not fit with the expectations of the future reader in 2023, especially health care professionals. The period of enthusiasm for the discovery of this tool has passed. I don't know if the data may be analyzed differently with some of the above questions. It would be important to be aware of this possibility because some answers of the present study may help to think about the future of type 1 diabetes mellitus, including the therapy of diabetes with device that is intended to automatically monitor glucose with algorithm, AI and machine learning with important ethical issues and limitations. 4-In addition, conclusions obtained during the pandemic period cannot be generalized since the expectations of health care professionals have now evolved. It is not sure that "Mapping our data onto the NPT concepts to understand how staff integrated Libre into routine care especially during a pandemic was a helpful approach". On the contrary I think that the covid pandemic creates confusion in the interpretation of the results. The data obtained with the Free Style libre sensor can be analyzed online. This is a great success of such device especially in the pandemic period. But now, we observed that data analysis is time consuming considering the generalized use of the sensors. Thus, the facility to get data online is now counterbalanced by the time required for the analysis of the data that possibly limits the time allocated to the face-to-face routine care with patients. In conclusion, this paper is interesting with an adequate methodology. Beyond the enthusiasm for the use of this sensor, the paper did not provide sufficient novelty in this topic considering the available literature.
--	---

VERSION 1 – AUTHOR RESPONSE

Reviewer: 1 Dr. Tatsuhiko Urakami, Nihon University School of Medicine Graduate School of Medicine	
1. Reviewer: 1 Dr. Tatsuhiko Urakami, Nihon University School of Medicine Graduate School of Medicine First, the participants of interview were few, and I consider the analyses of the results were incomplete to draw the conclusion: i.e., this study included only ten young people, and their parents and caregivers.	The study consists of 34 interviews with children and young people (aged 8-18) with type 1 diabetes (n=10), parents and carers of children aged 5-18 (n=14), and health professionals in direct clinical care of children/young people with type 1 diabetes. 12 families and 10 health professionals took part in 34 interviews. We believe this number is more than adequate for qualitative research methodology as no new themes were emerging by completion of these 34 interviews. We would like to draw attention to comments received from Reviewer 2 which state "Using this classical methodology in the

	field of social sciences, the number of participants is usually small and depends on the number of the themes explored in the study.” And “The methodology is well described and appropriate for the present study and reached the high level of quality in research in social sciences (or social psychology). There is no ethical issue.”
2. Reviewer: 1 Dr. Tatsuhiko Urakami, Nihon University School of Medicine Graduate School of Medicine Second, the results may be different between the age-groups. Therefore, the results should be compared between the age-groups, 8-12 year- and 13-18 year-group. However, I understand the comparison seems difficult because the number of patients in each group was too small for the statistical analysis.	Statistical analysis plays no real part in qualitative study methodology or reporting. We direct reviewers’ to the quantitative findings published from the grant in the following: [PDF] Implementation of flash glucose monitoring in four paediatric diabetes clinics, controlled before and after study to produce real world evidence of patient benefit. Study statistical and health economic analysis plan. Semantic Scholar Implementation of flash glucose monitoring in four paediatric diabetes clinics before and after study to produce real world evidence of patient benefit - NIHR Funding and Awards We would like to note that we were not seeking to compare the experiences of younger children with older teenagers, but report on experiences of using and overseeing patients using a Libre device. We would like to draw attention to Table 2, (Interview quotes from young people and parents organised by theme) these quotes highlight commonalities and differences in the experience of younger children (8-12yrs), teenagers (13-18yrs) and their parents e.g. an example of circumstances where parent/child views in relation to Libre differed e.g. “From a carer’s point of view they [alarms] are reassuring, from a younger child point of view they are incredibly annoying” #103 parent, child 8-12yrs, Libre 2 However, we have clarified the following points in the main text to further highlight commonalities and differences between age groups:

	pg.10 Libre 2 allowed them to carry on with activities including sport at school or paid work using the alarm or vibrate function to alert them to out of range glucose levels For some young people (particularly those aged 8-12) there was still the issue of the sensor being visible on their arm and questions from others caused them to hide it under clothes by wearing long sleeves.
3. Reviewer: 1 Dr. Tatsuhiko Urakami, Nihon University School of Medicine Graduate School of Medicine Third, the patient background was insufficiently described. Age at diagnosis, duration of diabetes, insulin treatment (MDI or CSII), HbA1c level, and serum C-peptide level in the patients should be described.	Because this is a qualitative study with a sample size of 12 families, and the study aimed to explore experiences of young people using Freestyle Libre sensors and their parents, (and to identify benefits and challenges to NHS staff of its adoption in their care provision) it would not be ethically appropriate to collect data which would not be meaningfully quantifiable. We do not believe the addition of such data would address the aims of the current study. Neither would it have been practically feasible to collect detailed medical data from social media volunteers (relating to multiple daily injections, continuous subcutaneous insulin infusion, HbA1c level, and serum C-peptide level) since we would have relied upon young person or parent report, we did not have access to patient medical records.
4. Reviewer: 1 Dr. Tatsuhiko Urakami, Nihon University School of Medicine Graduate School of Medicine Fourth, there is some difference between Libre 1 and Libre 2: i.e., Libre 2 has more benefit such as having hypo-/hyperglycemia alert system. The accuracy, particularly in low glucose level, is superior in Libre 2. Therefore, there may be difference of the results between the users with Libre 1 and those with Libre 2. The results in the study may be different according to use of Libre 1 and that with Libre 2.	We appreciate there are differences between Libre 1 and Libre 2, and therefore included information on the numbers using old or new devices on pg.7: Eight young people were using Libre 2, four were using Libre 1, one had stopped using Libre 1, and another child not interviewed had switched to a continuous blood glucose monitor (CGM). We also believe that we have conveyed participants' lived experiences in relation to whichever model they were using/willing to provide feedback about:

	pg.10 Participants also reported benefit at night such as being woken (by the Libre 2 alarm) and being able to treat a hypoglycaemic episode easily and sooner which resulted in them not feeling so terrible when they woke in the morning. Libre 2 allowed them to carry on with activities including sport at school or work using the alarm or vibrate function to alert them to out of range glucose levels (quote 2.1.2). pg.11-12 Parents liked the fact that they could scan their child at night with Libre 1 or use the alarm with Libre 2. Models. Once again, we do feel we need to comment that whilst there may be perspectives that are different between Libre 1 and 2, which we have included, the purpose of this report is not to compare quantitative data such as number of hypoglycaemic events using Libre 1 compared to Libre 2.
5. Reviewer: 1 Dr. Tatsuhiko Urakami, Nihon University School of Medicine Graduate School of Medicine Fifth, interpretation of the results was complicated in the study. The authors should summarize the results easy to understand.	As mentioned previously – we would like to draw attention to comments received from Reviewer 2 which state: “The methodology is well described and appropriate for the present study and reached the high level of quality in research in social sciences (or social psychology). There is no ethical issue.”
6. Reviewer: 1, Dr. Tatsuhiko Urakami, Nihon University School of Medicine Graduate School of Medicine Sixth, the usefulness of real-time CGM for children with type 1 diabetes should be mentioned in the discussion.	We do mention that: Flash glucose monitoring, and other CGM devices, have the potential to improve long-term glycaemic control. Young people should be supported and encouraged to use the sensors effectively. In a qualitative study exploring family and clinician perspectives of libre, we feel this is sufficient to acknowledge the benefits of real time glucose measurement.
7. Reviewer: 1 Dr. Tatsuhiko Urakami, Nihon University School of Medicine Graduate School of Medicine Finally, the references lack important studies described the clinical use of FGM in children and adolescents with type 1 diabetes. The ISPAD Consensus Guideline	We agree and have referenced the suggested consensus guidelines as pointed out by reviewer 1:

2022 is at least necessary for the discussion. Recent other studies should be included in the references.	Flash glucose monitoring, and other CGM devices, have the potential to improve long-term glycaemic control.^{13,14} Young people should be supported and encouraged to use the sensors effectively.^{7,13,14} 13. Craig ME, Codner E, Mahmud FH, Marcovecchio ML, DiMeglio LA, Priyambada L et al. ISPAD Clinical Practice Consensus Guidelines 2022: Editorial. Pediatr Diabetes. 2022; 23; 1157-59. 14. De Bock M, Codner E, Craig ME, Huynh T, Maahs DM, Mahmud FH et al. International Society for Pediatric and Adolescent Diabetes (ISPAD). ISPAD Clinical Practice Consensus Guidelines, 2022: Glycemic targets and glucose monitoring for children, adolescents and young people with diabetes. Pediatr Diabetes. 2022; 23; 1270-1276.
Reviewer: 2 Prof. FABRIZIO ANDREELLI, Hôpital Pitié-Salpêtrière Service Néphrologie ” The methodology is well described and appropriate for the present study and reached the high level of quality in research in social sciences (or social psychology). There is no ethical issue. In conclusion, this paper is interesting with an adequate methodology. Beyond the enthusiasm for the use of this sensor, the paper did not provide sufficient novelty in this topic considering the available literature.	
8. Reviewer: 2 Prof. FABRIZIO ANDREELLI, Hôpital Pitié-Salpêtrière Service Néphrologie Major concerns 1-The description of the results (on the one hand for the patients and their parents and on the other hand for the healthcare professionals) summarized adequately the most important messages from the interviews and also the constructs from the health professional data. But Freestyle Libre sensors are used since 2017 in many countries in Europe. Individual interviews and focus groups in patients with type 1 diabetes and their parents in our routine clinical care, confirmed all the data presented in the paper of Lucy Beasant and colleagues especially (some interesting sentences from the discussion)  - Life was much easier and better for young people freed from the burden of frequent finger-pricking - Changing to flash glucose monitoring gave 	Thank you for this feedback. We appreciate that 2 qualitative studies (Boucher et al 2020) have been published previously by authors in New Zealand, but these were specific to parents of adolescents and young adults (aged 14-19 years) with type 1 diabetes who were either not meeting glycaemic targets or considered at high-risk glycaemic control soon after starting flash glucose monitoring. Limitation of the Boucher et al study included that findings ‘may not be generalizable to longer periods of use’ and the ‘sample may be over representative of those who had a positive experience with flash glucose monitoring’ since their sample did not include those who had stopped using Libre. We appreciate that our comparison with past relevant literature in the discussion may have

them confidence and increased independence in managing their condition

- Parents' quality of life also improved, and they appreciated the access to real time data.

-Health professionals reported that they were very enthusiastic about flash glucose monitoring and were coping with the extra burden on their time and data overload.

Similar conclusion has already been published since 2017 and the references of these papers are listed in the manuscript of Lucy Beasant and colleagues. Thus, the discussion did not provide comparison with the literature and failed to demonstrate sufficient novelty in this field.

been lacking. We believe we have added to available knowledge base from a lived perspective. Our sample of participants included those with T1D aged 8-17 years, and were a mix of those who had recently started using flash glucose monitoring tech, those who had stopped using it, and those who had been using it for some time in England.

We have therefore added more detail to demonstrate how our study adds to existing findings in this area and is sufficiently novel in the field to warrant publication:

Our findings concur with these recently published studies, but importantly we have added to the available knowledge base by recruiting those who had recently started, stopped, or had been using Libre for some time in England. Our novel finding relating to the 'visibility' of Libre and the potential for young people to realise the potential for positive acknowledgement from others, or intervention from the wider public should the wearer have a hypoglycaemic episode add to existing evidence in this area that reports visibility to others as a moderately negative feature of the device.⁶ Parents involved in our study were often very tech savvy, and the 'tech journey' they described highlights the potential for further health inequalities as diabetes technology quickly changes, resulting in families on low incomes not being able to access new technology until it is funded on the NHS.

We would like to draw attention to pg.12 'Co-produced animations', as we believe it is particularly important to publish findings in relation to the two short animations (narrated by young people) which were developed for children and teenagers using direct quotes from the findings which are hosted on Digibete (digibete.org.uk) and the NIHR Bristol Biomedical Research Centre website: <https://www.bristolbrc.nihr.ac.uk/FLASH-animations>

	We also highlight that our findings report contemporaneous views of health professionals, alongside those of young people and their parents.
9. Reviewer: 2 Prof. FABRIZIO ANDREELLI, Hôpital Pitié-Salpêtrière Service Néphrologie Major concerns 2- In the discussion, the following sentence “Manufacturers and clinicians should provide good education on interpreting glucose data and support making informed treatment decisions to maximize the benefits of this technology” is strange because the use of Free Style libre sensors in adults and in children with type 1 diabetes is a part of our routine care including the analysis of the data by the patients and their parents. This sentence suggested that this sensor is a new device that is currently being evaluated. And this is not the case.	Thank you for this feedback. Although Free Style Libre sensors are now widely used in routine clinical care, families felt that some information (e.g. initial Abbott training package in terms of content and delivery) was not tailored to children and young people, we have amended this sentence in the discussion so that this point is more clear: Our findings suggest that manufacturers should provide tailored education that is more accessible to children and young people, and clinicians should continue to support families to make informed treatment decisions to maximize the benefits of this technology. Also see pg.15 webinars are still largely ‘adult centric’ and not very suitable for young people
10. Reviewer: 2 Prof. FABRIZIO ANDREELLI, Hôpital Pitié-Salpêtrière Service Néphrologie Major concerns 3-Beyond the initial analysis of the data, the similarity or difference of the themes and constructs addressed by each category of participants (patients, parents, health care professionals) should be the most interesting conclusion of such study. The discussion does not sufficiently highlight the differences in participants' expectations. For example, some children and adults refuse to use this sensor or disconnect the alarms despite the improvement of long-term glycemic control established in clinical studies. Their decision is in contrast with the expectations of their parents. How can we explain these differences? How each category considers type 1 diabetes mellitus: a jail for young patients ? A challenge for parents to succeed ? A new data collection like the “temperature curve” or “equivalent to accumulated biological data over time such as kidney or liver function, needing mathematical analysis with means and data distribution but far away the specific history of each patient for health care professionals”? Can we develop some specific strategy to convince the patients to use these sensors? In this strategy of permanent glucose monitoring, to what extent parents and professionals may accept mistakes	Thank you for these detailed comments. We have addressed the following points: The discussion does not sufficiently highlight the differences in participants' expectations... Their decision is in contrast with the expectations of their parents. We have added the following point to the discussion: Conflicts between parents and children occurred but were largely overcome, with the majority of

of patients in their routine management of the disease ? On the contrary, thinking that everybody can see your mistakes may be stressful for the patient community. Is it a sufficient reason to reject the use of the device? Yes, if we consider that parents and children share the same data in real life: less stress for parents; more discussion and sometimes conflicts for the young patients with their parents. This sentence of the discussion “Young people should be supported and encouraged to use the sensors effectively” is key and should be developed considering the data available in the literature and also the analysis of the limitations mentioned by the patients. In other words, such device has first to help patients and the discussion seems less patients-centered than one could expect. In addition, as mentioned above, the results of the present study sometimes did not fit with the expectations of the future reader in 2023, especially health care professionals. The period of enthusiasm for the discovery of this tool has passed.	young people still choosing to use Libre because it resulted in them having to finger prick less often. Can we develop some specific strategy to convince the patients to use these sensors? This was outside the scope of our study and may be for future research. thinking that everybody can see your mistakes may be stressful for the patient community. We highlight this on pg11: Finger pricking enabled a child not to think about their diabetes as much, but using Libre and having access to continuous data could be overwhelming or demoralising particularly when their management was poor (quote 2.2.4). Young people should be supported and encouraged to use the sensors effectively” is key... As mentioned above, we have amended this sentence in the discussion to highlight that children and young people need to be better supported: Our findings suggest that manufacturers should provide tailored education that is more accessible to children and young people, and clinicians should continue to support families to make informed treatment decisions to maximize the benefits of this technology.
--	---

	the discussion seems less patient-centered than one could expect We draw attention the following points in the discussion: Implications for practice include observations that using Freestyle Libre sensors improves the consultation process as visible data on a clinic computer allow collective visualisation of good and less good control and a collaborative process of adjustments to be made which facilitates greater patient/family involvement in their care.
11. Reviewer: 2 Prof. FABRIZIO ANDREELLI, Hôpital Pitié-Salpêtrière Service Néphrologie Major concerns I don't know if the data may be analyzed differently with some of the above questions. It would be important to be aware of this possibility because some answers of the present study may help to think about the future of type 1 diabetes mellitus, including the therapy of diabetes with device that is intended to automatically monitor glucose with algorithm, AI and machine learning with important ethical issues and limitations.	The reviewer makes a very important point about the potential of a functioning closed-loop, insulin/CGM/machine learning system and how young people may rebel against such a system to their eventual disadvantage. However, due to a-priori topic guide question development, we cannot re-analyse the data for such information as this was not sought in the confines of an assessment of current libre models, important as it will be for future diabetes research. We have therefore added the following to pg.16 to highlight that health professionals in the current study were enthusiastic about engaging with tech developments in the future: Health professionals reported that they were very enthusiastic about flash glucose monitoring, were coping with the extra burden on their time and data overload, and were keen to continue using new and improved Libre models in future.
12. Reviewer: 2 Prof. FABRIZIO ANDREELLI, Hôpital Pitié-Salpêtrière Service Néphrologie Major concerns 4-In addition, conclusions obtained during the pandemic period cannot be generalized since the expectations of health care professionals have now evolved. It is not sure that "Mapping our data onto the NPT concepts to understand how staff integrated Libre into routine care especially during a	Thank you for making these points. In response to: 'conclusions obtained during the pandemic period cannot be generalized since the expectations of health care professionals have now evolved'

pandemic was a helpful approach". On the contrary I think that the covid pandemic creates confusion in the interpretation of the results. The data obtained with the Free Style libre sensor can be analyzed online. This is a great success of such device especially in the pandemic period. But now, we observed that data analysis is time consuming considering the generalized use of the sensors. Thus, the facility to get data online is now counterbalanced by the time required for the analysis of the data that possibly limits the time allocated to the face-to-face routine care with patients.	Although it is not possible to generalise from qualitative findings, we do believe that the COVID-19 pandemic has led to changes in the way NHS appointments and technology training continue to be delivered 'post pandemic' See pg.15: When reflecting on the introduction of flash glucose monitoring devices, health professionals highlighted some interesting points. Using online training has become more acceptable since the COVID-19 pandemic lockdowns, In response to: 'This is a great success of such device especially in the pandemic period. But now, we observed that data analysis is time consuming considering the generalized use of the sensors. Thus, the facility to get data online is now counterbalanced by the time required for the analysis of the data that possibly limits the time allocated to the face-to-face routine care with patients.' In our interviews with clinical staff this was touched on, and we say in discussion (pg.16): Health professionals reported that they were very enthusiastic about flash glucose monitoring and felt they were coping with the extra burden on their time and data overload.' In conclusion (pg.17) we say: 'Health professionals were positive about its use, reporting practical, clinical and educational benefits for patients, whilst also identifying the increased time commitments for individual family support within limited health care resources.' To reflect the reviewer's very accurate appraisal of this situation we have added the following to the final section of the conclusions:
--	--

	With the rapid adoption of some form of CGM for all children with diabetes and evolving intricacies of insulin delivery, it may be the case that reviewing downloads regularly and the additional time to analyse at each clinic visit, may eventually reduce the number of face-to-face appointments available for each child and family due to limited staffing resources.
--	--